# Pandemic catch-22: The role of mobility restrictions and institutional inequalities in halting the spread of COVID-19

**Adnan M. S. Fakir**[1,2]*, **Tushar Bharati**[1]

**1** Economics Department, University of Western Australia Business School, Perth, Australia, **2** Economics and Social Sciences, BRAC University, Dhaka, Bangladesh

* adnan.fakir@uwa.edu.au

## Abstract

Countries across the world responded to the COVID-19 pandemic with what might well be the set of biggest state-led mobility and activity restrictions in the history of humankind. But how effective were these measures across countries? Compared to multiple recent studies that document an association between such restrictions and the control of the contagion, we use an instrumental variable approach to estimate the causal effect of these restrictions on mobility, and the growth rate of confirmed cases and deaths during the first wave of the pandemic. Using the level of stringency in the rest of the world to predict the level of stringency of the restriction measures in a country, we show while stricter contemporaneous measures affected mobility, stringency in seven to fourteen days mattered most for containing the contagion. Heterogeneity analysis, by various institutional inequalities, reveals that even though the restrictions reduced mobility more in relatively less-developed countries, the causal effect of a reduction in mobility was higher in more developed countries. We propose several explanations. Our results highlight the need to complement mobility and activity restrictions with other health and information measures, especially in less-developed countries, to combat the COVID-19 pandemic effectively.

## 1 Introduction

By December 31, 2020, the COVID-19 pandemic had infected over 90 million people and claimed almost 2 million lives. Countries across the world have responded with what might well be the set of biggest state-led mobility and activity restrictions in the history of humankind. The hope is to contain the contagion and reduce the congestion in health-care utilization. But besides being controversial and costly, such measures may not always be successful in containing the spread and can, sometimes, worsen the situation (see, among others, [1–10]). The trade-offs are bigger for developing countries. In the absence of proper social security support, they face a catch-22 situation where strict mobility and activity restrictions, especially if ineffective, will unnecessarily increase the economic cost through lost livelihoods and, perhaps, even compromise the immunity of the vulnerable population. The natural question that then

**Data Availability Statement:** 1) Oxford Government Response Tracker: https://www.bsg.ox.ac.uk/research/research-projects/coronavirus-government-response-tracker 2) Google

Community Mobility Report: https://www.google.com/covid19/mobility/ 3) Our World In Data COVID-19 Testing: https://ourworldindata.org/coronavirus-testing 4) Johns Hopkins Center for Systems Science and Engineering COVID Tracking Data: https://coronavirus.jhu.edu/ 5) World Development Indicators: https://databank.worldbank.org/source/world-development-indicators 6) OECD, Eurostat, Hospital Beds per 100k Population: https://data.oecd.org/healtheqt/hospital-beds.htm 7) United Nations Statistics, Handwashing Facilities: https://unstats.un.org/sdgs/unct-toolkit/data-resources/ 8) Global Burden of Disease Study 2017 Results, CVD Death Rate: http://ghdx.healthdata.org/gbd-2017 9) Economist Intelligence Unit, Democracy Index: https://www.eiu.com/topic/democracy-index 10) Transparency International, Corruption Perception Index: https://www.transparency.org/en/cpi/2019.

**Funding:** The author(s) received no specific funding for this work.

**Competing interests:** The authors have declared that no competing interests exist.

follows is whether such measures have been effective in controlling the COVID-19 contagion. If yes, what institutional factors contribute to their effectiveness?

Multiple recent studies have submitted that there exists a negative association between such restrictions and the contagion [11–17]. However, much of the work either simulate counterfactual scenarios or documents association between the restrictions and the contagion. With studies suggesting a steep economic cost of such restrictions, to be able to design optimal mitigation policy for COVID-19 and future pandemics, it is crucial to understand whether, when, where, and how much do these restrictions causally affect the contagion [18]. Few studies attempt to identify causal effects of the restrictions using a difference-in-differences (DiD) design—comparing regions with high and low levels of restrictions [12, 16, 17]. But the restrictions were almost always in response to the disease situations in the region. Areas with worse contagion or more watchful populations might have enacted stringent restrictions relatively early. Since these factors could have also affected the evolution of the disease scenario, the assumption of parallel trends underlying the DiD methodology is unlikely to hold.

We propose an instrumental variable approach to estimate the causal effect that the level of stringency of the restrictions had on human mobility and the growth rate of the contagion. In deciding whether to impose restrictions, national and local governments took into account not only the prevailing disease situation in the country (the factor that confounds DiD estimates) but also what they expected would happen in the future in the presence and absence of such restrictions. Lacking perfect foresight, they made predictions based on their observations of the condition in the rest of the world [19]. Governments that witnessed a rapid increase in the number of COVID-19 cases and subsequent mobility and activity restrictions in the rest of the world in the days following the first confirmed case in their own country, sprung into action swiftly and imposed stricter restrictions. It is important to note that enacting strict policy measures does not necessarily translate to enforcement or compliance, but rather an acknowledgment of the need for stricter restrictions, and possibly an intent. Building on this insight, we use the day-to-day changes in the stringency of the restrictions in the rest of the world to instrument how stringent a country's internal mobility and activity restrictions were.

We conduct our analysis combining high-frequency measures of mobility data from Google's daily mobility reports, country-date-level information on the stringency of restrictions in response to the pandemic from Oxford's Coronavirus Government Response Tracker (OxCGRT), and daily data on people tested, confirmed cases, and deaths attributed to COVID-19 from Our World In Data and the Johns Hopkins Center for Systems Science and Engineering (CSSE). Using the instrumental variable technique, we estimate large causal effects of stricter restrictions on mobility and the weekly growth rate of recorded cases and deaths attributed to COVID-19. In comparison, we find that more stringent restrictions have weak marginal effects on the growth rate of tests conducted. Consistent with the current scientific understanding that an infected human can infect another human up to 14 days since being infected, we see that the level of stringency of the restrictions in the previous two weeks matters more than the contemporaneous level or level of restrictions 3 weeks in the past (see Qiu et al. (2020) [15] and the studies they cite for a discussion of the incubation and the infection period). We also document considerable differences between the correlation and causal estimates that raise concerns over the use of the association estimates from previous studies to evaluate the costs and benefits of the restrictions.

Next, we show that the effectiveness of the restrictions varies significantly across countries as per their institutional capacities. In particular, more stringent measures help more in richer, more educated, more democratic, and less corrupt countries with older, healthier populations and more effective governments. Finally, we draw attention to the observation that announcing restrictions does not necessarily imply a reduction in mobility; it depends on the level of

compliance and enactment of the policies. The estimated reduced-form effects of stringency on the growth rate of cases and deaths incorporates the differential compliance across countries. Utilizing a recursive mixed-process model, we show that even though the stricter restrictions had a larger negative marginal effect on mobility in relatively less-developed countries, they were more effective in containing the contagion in more developed countries. However, as readers will see, while this result is stark when the outcome variable is cases to tests ratio, it is not as stark for deaths to tests ratio. Consistent with the institutional heterogeneity results, these results indicate that imposing mobility restrictions is not enough to contain the contagion in developing countries, and the benefits reaped from high stringency are lower relative to developed nations. The restrictions should be effectively complemented with other policy measures, such as raising awareness about best practices when these restrictions are imposed, and health and economic assistance for those affected [20, 21].

The findings have important policy implications. COVID-19 is not the first and will not be the last epidemic to afflict humanity. Better future preparedness requires a better understanding of when and how to act in times of such crises. Understanding the effectiveness of mobility and activity restrictions in containing contagions will not only help us optimize our current response to COVID-19 but also prepare us better to face future disease outbreaks. The institutional heterogeneity analysis suggests that increasing stringency alone might not be enough, especially in developing countries where labor market conditions, lacking health infrastructure, and constraints on implementation infrastructure might limit the effectiveness of these restrictions. Since the economic downturn can negatively affect the health and welfare outcomes in poorer countries more than in rich countries where the transition into work from home is relatively easier, this raises serious concerns about the cost-effectiveness of stringent mobility and activity restrictions in the absence of complementary policies. The results call for a country-specific policy response suited to the institutional capacity and socio-economic circumstances of the country.

## 2 Data and empirical specification

### 2.1 Data and summary statistics

For our analyses, we collate and link country-level daily data from the following sources:

**2.1.1 Google community mobility reports.** To facilitate better monitoring and compliance to the nationwide and local lockdown decisions and social-distancing requirements to reduce the transmission of the COVID-19 contagion, Google has released publicly daily aggregated data on changes in mobility across six key high-level location categories in 131 countries. The mobility measures reflect how busy these places are. The six location categories are groceries and pharmacies, retail and recreation sites, parks, transit stations, workplaces, and residences. We source the mobility data from these reports from the 15th of February to the 30th of July reflecting daily percentage changes in reference to a baseline. The baseline is the median value of mobility for the corresponding day of the week during the 5 weeks of January 3, 2020, to February 6, 2020. These measures of changes in mobility across the six location categories serve as our first set of outcome variables.

While a reasonable measure of the extent of compliance to the restrictions, or successful enactment of the policies, the data comes with certain caveats. The reports are generated using a technology similar to the real-time anonymized Google Maps traffic data, and as such are reflective of only those users who have their location history setting turned on in their (cellphone, tablet, etc.) Google account [22]. Therefore, while the data is impressive, it is not representative of the population at large. Another important aspect to note is that while the residential category shows the relative change in daily time spent at home, the other measures

reflect respective daily relative changes in the number of individual visits. So, the residential category carries a different unit of measurement than the other categories and thus should be interpreted as such.

**2.1.2 Oxford COVID-19 Government Response Tracker (OxCGRT).**   OxCGRT provides a comprehensive and systematic country-level daily stringency index, constructed based on common policy responses implemented by governments to combat COVID-19 [23]. Stringency is measured as a composite score, equally weighted and normalized between 0 and 100 for each country (with 100 being the strictest response), using eight ordinal indicators of containment, movement restriction, and closure policies, and a ninth indicator measuring the coordinated presence of public awareness campaigns on the pandemic. The containment indicators include school closures, workplace closure, cancellation of public events, restrictions on gatherings, public transport closure, stay-at-home requirements, internal movement restrictions, and international travel controls.

Since the stringency index further tracks how quickly governments implemented or rolled out their policy measures, we use the index as our primary independent variable of interest. As contemporary stringency measures would affect mobility but its effect on the growth of the pandemic would be observable only days after, we also use lagged values of the index in our analyses. While the index provides a numerical score to the strictness of the policies enacted, it does not reflect the compliance or effectiveness of the stringency put in place. The case for compliance is more relevant when exploring the legally binding nature of the policies. For example, Katafuchi et al. (2020) [24] shows that even without the declaration of a state of emergency in Japan people partially suppressed their mobility. Although, expectedly, mobility was suppressed more with the state of emergency in place. This paper aims to provide a more comprehensive overview of mobility changes, and their subsequent role in containing the contagion, with or without legally binding policies. Hence, while a higher score in the index reflects a willingness for greater stringency, it does not necessarily translate to a country's response being better than countries with a lower score.

It is also important to note while some countries enacted rigid mobility and activity restrictions, other countries adopted more flexible measures. Further, these levels of flexibility/rigidity have changed within a country over time. OxCGRT integrates these fluctuations into their stringency index by categorizing each of the nine indicators into ordinal levels by the rigidity of the restriction. For example, school closures are categorized into "0—no measures; 1—recommend closing or all schools open with alterations resulting in significant differences compared to non-Covid-19 operations; 2—require closing (only some levels or categories, for eg. just high school, or just public schools); 3—require closing all levels" [25]. The final stringency index is then a composite weighted index where higher values reflect the levels of rigidity of the restrictions. Please refer to [25] for details on the index's construction. S1–S9 Figs in S1 Appendix provide event graphs of the stringency index by country over time for all 127 countries in our sample. Values above 50 can be interpreted as the country undertaking relatively stricter measures.

**2.1.3 COVID-19 outbreak data.**   We source COVID-19 country-specific daily data on confirmed cases per million and deaths attributed to COVID-19 per million from the Johns Hopkins Center for Systems Science and Engineering (CSSE) COVID-19 data repository [26]. We combine this with tests per million population data collated by Our World In Data (OWID). Since it takes some time for delayed reporting to be reflected in the dataset, we use 7-day moving averages of the outbreak variables and restrict our focus to events between February 15, 2020, to July 30, 2020. It is perhaps worth mentioning that even if we do not use moving averages, our conclusions remain the same. Results are available upon request. We construct daily growth rates of 7-day moving averages for the outbreak variables—tests, cases,

cases to tests ratio, deaths, and deaths to tests ratio—and use them as our second set of outcomes. We explain the rationale behind using ratios in Section 2.2. OWID collects testing data from country-specific official government reports and is available only from 85 countries. We limit our analysis to the 127 countries (80 for tests) that have mobility, stringency, cases, and deaths data available. The countries are listed in S1 Table in S1 Appendix.

Several studies and media outlets have reported that due to country-specific differences in testing rates, data aggregation, and reporting quality, the number of cases and deaths are under-reported [27–29]. Testing data, when available, has a strong selection bias with many countries screening and testing only those people who presented symptoms. The extent of this selection bias might be systematically related to country-specific characteristics. While we control for country and time fixed effects in our empirical specifications, it will not account for systematic changes in selection bias over time across countries. Therefore, this study, like all studies utilizing the CSSE and the OWID data, should be interpreted with a healthy dose of skepticism. We further elaborate on such data limitations in Section 2.2 and what we do to best circumvent the constraints.

**2.1.4 Heterogeneity variables.** To investigate the role of institutional heterogeneity in the impact of the restrictions across developing and developed countries, we link our data with various pre-COVID-19 country-specific demographic, health, and governance factors, that may aid or hinder the stringency effect on people's mobility and the spread of the disease. Along the demography dimension, we examine heterogeneity by population density, education, poverty headcount, economic inequality (Gini index), the share of the population aged 65 years or above, and air pollution per capita (measured by the concentration of suspended particulate matter in the air with a diameter of 2.5 micrometers or less—PM2.5). We also examine heterogeneity by available hospital beds per 100 thousand population (a proxy of available healthcare infrastructure), the share of the population with hand-washing facilities on-premises (a proxy for the availability of tools to combating the growth in transmissions), and the death rate from cardiovascular diseases (CVD) (a proxy for share of the immune-compromised population who face higher risks from COVID-19).

Finally, we examine heterogeneity along country's governance indicators using the Economist Intelligence Unit (EIU) democracy index, government effectiveness from the Worldwide Governance Indicators (WGI) [30], and the corruption perception index (CPI) developed by Transparency International (TI), where larger values represent cleaner countries. The vast majority of data from the demography and health dimensions are sourced from the World Development Indicators (WDI), United Nations Population Division, or the Global Burden of Disease Collaboration Network. S2 Table in S1 Appendix provides details of the sources for each of the variables used, and Table 1 below presents the summary statistics.

The stringency index appears to be skewed to the left with a mean value of 64 below the median of 71, meaning there is a relatively long tail of days with lower stringency scores. All mobility measures, excluding residential mobility, show a percentage decrease in the visits with the decrease being greatest at about 38 percent at transit stations, followed closely by mobility around retail and recreation sites. On the other hand, the percentage change in time spent at home increases by about 13 percent. Segregating the measures by developing vis-à-vis developed countries, reported in S3 Table in S1 Appendix, shows that even though mean stringency is relatively similar in both cohorts, mobility changes were mostly greater in developed vis-à-vis developing countries (except in mobility around parks, and slightly for grocery and pharmacy). This could be an initial indication of either overall lower compliance to mobility restrictions in developing countries, or greater self-regulation in developed countries (for example, while transit stations see a decrease of 34 percent in developing countries, developed countries see a 40 percent decrease).

**Table 1. Summary statistics.**

|  | N | Mean | SD | Median | Min | Max |
|---|---|---|---|---|---|---|
| **Oxford Government Response Tracker** | | | | | | |
| Stringency Index | 18322 | 64.70 | 24.24 | 71.30 | 0.00 | 100.00 |
| **Google Mobility Measures** | | | | | | |
| Retail and Recreation (% change) | 18322 | -33.34 | 26.54 | -30.00 | -97.00 | 42.00 |
| Grocery and Pharmacy (% change) | 18317 | -15.07 | 22.39 | -11.00 | -97.00 | 94.00 |
| Parks (% change) | 18321 | -7.38 | 53.33 | -16.00 | -95.00 | 517.00 |
| Transit Stations (% change) | 18322 | -38.39 | 24.53 | -39.00 | -95.00 | 39.00 |
| Workplaces (% change) | 18322 | -27.22 | 22.80 | -26.00 | -92.00 | 80.00 |
| Residential (% change) | 18246 | 13.14 | 10.16 | 12.00 | -16.00 | 55.00 |
| **Outbreak Variables** (Growth Rates of 7-Day Moving Average per million population) | | | | | | |
| Tests | 10660 | 0.05 | 0.06 | 0.03 | -0.01 | 0.97 |
| Cases | 18322 | 0.07 | 0.13 | 0.03 | -0.13 | 2.65 |
| Cases to Tests | 10660 | 0.01 | 0.06 | -0.00 | -0.49 | 1.53 |
| Deaths | 15269 | 0.07 | 0.17 | 0.02 | -0.01 | 4.01 |
| Deaths to Tests | 9676 | 0.02 | 0.10 | -0.00 | -0.44 | 3.24 |
| Days since first case (by Country) | 18322 | 76.99 | 44.80 | 76.00 | -76.00 | 194.00 |
| **Heterogeneity Variables** | | | | | | |
| Population Density | 18158 | 232.08 | 772.33 | 87.32 | 1.98 | 7915.73 |
| Primary Education | 11557 | 78.79 | 23.23 | 87.54 | 13.87 | 100.00 |
| Poverty Headcount (2011 PPP) | 16294 | 9.43 | 16.02 | 1.40 | 0.00 | 62.90 |
| Gini Index | 16152 | 37.87 | 7.93 | 36.40 | 24.20 | 63.00 |
| Population Aged 65 or older | 18158 | 10.01 | 6.65 | 7.65 | 1.14 | 27.05 |
| PM2.5 (2010–2017 Average) | 17219 | 28.25 | 19.93 | 22.64 | 6.46 | 98.25 |
| Corruption Perception Index | 17373 | 47.10 | 19.70 | 41.00 | 14.00 | 88.00 |
| Democracy Score | 17412 | 6.01 | 2.09 | 6.33 | 1.93 | 9.87 |
| Governance Effectiveness | 18196 | 0.21 | 0.96 | 0.11 | -2.24 | 2.23 |
| Hospital Beds per 100k Population | 16936 | 3.17 | 2.60 | 2.40 | 0.10 | 13.05 |
| Handwashing Facilities | 8032 | 57.15 | 29.72 | 59.61 | 2.73 | 99.00 |
| CVD Death Rate | 18185 | 236.83 | 117.38 | 218.61 | 79.37 | 597.03 |

Using available data from 15 Feb to 30 July, 2020.

Summary statistics by level of country development are available in S3 Table in S1 Appendix

Mean cumulative 7-day moving average daily growth rates of tests, cases, and deaths are 5, 7, and 7 percent, respectively, while that of the ratios are smaller, with cases to tests ratio at 1 percent and deaths to tests ratio at 2 percent. Finally, while the mean statistic of the variables provides a snapshot of the overall sample, developing countries are, on average, significantly less educated, poorer, younger, more polluted, lack adequate health infrastructure, face greater corruption, and have poorer levels of democracy and government effectiveness (see S3 Table in S1 Appendix).

## 2.2 Data limitations

In the absence of a unified framework for testing and reporting for COVID-19 infections, the available data suffers from a multitude of problems. These range from no count of COVID-positive people who are not diagnosed including those asymptomatic, varying assay specificity and sensitivity leading to false-negatives or false-positives, differences in testing, comorbidities,

imperfect reporting, the release of incorrect data, and delays in reporting, to name a few. Ange-lopoulos et al. (2020) [31] discusses how the problems could bias estimation in either direction depending on their relative magnitude, and Millimet and Parmeter (2019) [32] provides a discussion of cases when data is skewed in one direction due to one-sided measurement errors. While many of these data issues cannot be resolved, we interpret our estimates with caution and perform various robustness checks to minimize the bias in the comparisons we make.

One glaring problem is the misreporting of the number of (per capita) confirmed cases. To account for delays in reporting, we use 7-day moving averages of COVID-related measures. However, the number of confirmed cases depends on the number of tests conducted, which itself varies across time within each country. This variation across time within each country will not be absorbed by the country or time fixed effects that we include in our specifications. In general, however, countries have increased testing over time (which could be with country income level or other country characteristics), albeit at differential rates. Using the growth rate of cases would, therefore, bias our results. Instead, we use the ratio of cases to tests. If the rates of infection in the untested population are similar to the rates of infection in the tested population, the ratio of cases to tests is a better reflection of the prevailing disease situation.

Similar measurement errors plague the information about deaths due to COVID-19. For surveillance purposes, WHO [33] defines a COVID-19 death "as a death resulting from a clinically compatible illness in a probable or confirmed COVID-19 case, unless there is a clear alternative cause of death that cannot be related to COVID-19 disease." Since, COVID deaths are ascertained in relation to probable or confirmed COVID-19 cases, where the latter depends on the number of tests within a country, data about deaths due to COVID from different countries suffer from different levels of measurement errors. For example, even within Europe, countries like Belgium have a more comprehensive approach to reporting deaths due to COVID than the United Kingdom that does not count non-hospital fatalities [34]. But if the rates of recovery from infections are similar in the tested and the untested population, the infection fatality ratio ($IFR = \frac{deaths}{infected}$) and the case fatality ratio ($CFR = \frac{deaths}{cases}$), reported by many countries, can be relatively good measures of COVID-related mortality in the population. However, as WHO [35] points out, the accuracy of these measures relies on two assumptions. First, that the likelihood of detection of confirmed cases and deaths due to COVID is consistent over time; and second, all detected cases are resolved (either recovered or died). Given testing and reporting limitations, often both these assumptions might be violated.

Other studies have explored statistical methods to correct for the bias, each with their own caveats (see for example, [36–39]). However, there is no agreement on whether these statistical measures yield better estimates and whether one measure is superior to the others. Instead, we construct a deaths to tests ratio as a measure of the disease situation. We prefer using deaths to test ratio to deaths to infected and deaths to cases ratio because the latter measures have time-varying measurement error in both the numerator and the denominator. Without any information on the relative degree of measurement error in the numerator and the denominator, it would be difficult for us to sign the bias. While the information on tests conducted is not free of measurement error, unless deliberately misreported by the reporting country, the error should be relatively small. So, while the deaths to tests ratios are not free of error, it is, arguably, the most comparable measure across countries. The estimated effects, albeit differing in magnitude, remain qualitatively similar when we use other measures: death per 100,000, infection fatality ratio, or case fatality ratio. In addition, since countries varied in their approaches to track the spread of the disease, this measure allows us to check the robustness of our results by limiting our sample to countries with relatively more reliable infection rate data.

There are three possible ways to limit the sample to countries that provide reliable test data. First, in the absence of testing a randomly selected sample from the population that most countries lack the resources to implement, Angelopoulos et al. (2020) [31] expands on how contact tracing can be a powerful tool that allows otherwise intractable biases to be controlled. Contact tracing expands to include a much larger section of the target population, specifically a larger portion of mild and asymptomatic cases, that are otherwise left out from the testing pool. The authors show that by assuming non-response rates to contact tracing as identical for asymptomatic and symptomatic cases, asymptotically unbiased estimations can be obtained. Therefore, we could limit our sample to the 59 countries that conducted comprehensive contact tracing and tests (which also has a good split of developed and developing countries, as indicated in S1 Table in S1 Appendix). S4 Table in S1 Appendix reports the mean of the 7-day moving average growth rates by no, limited, and full contact tracing. As expected, the growth rate means fall with increasing contact tracing. But the means fall only slightly when compared to the full sample, except for the cases to tests ratio.

A second approach could be to limit the sample by the country testing policy. Data from countries with an open public testing policy might be better representative than from countries with only limited testing. But it would still suffer from selection bias as people might self-select into testing. OxCGRT categorizes testing policy into four groups: (1) no testing policy, (2) limited testing of those who both have symptoms and meet specific criteria (eg. key workers, admitted to hospital, came into contact with a known case, returned from overseas), (3) symptom-based testing, and (4) open public testing. S4 Table in S1 Appendix once again shows the falling growth rate means with better testing policy. However, only 34 countries conducted open public testing within the timeframe of our study, resulting in a much smaller sample of 2,866 (compared to 10,539 for the full sample). Further, since only developed countries with adequate resources were able to adopt this testing approach, restricting the sample by the country's testing policy will be against the purpose of this study.

A third approach could be restricting the sample according to the type of testing data reported. A vast majority of countries report only the total number of tests conducted, double-counting follow-up or repeat tests for the same person [40]. This double-counting would, in most cases, exert an upward bias on the estimated effects. Thus, limiting the sample to only the countries that report the number of people tested may be a viable approach. Austin and Kachalia [2020] [40] posits that these countries may also be reporting quality data relative to others. However, only 21 countries reported the number of people tested, and when compared, S4 Table in S1 Appendix shows that the growth rate means are fairly similar between the two groups that reported the number of people tested and that reported the total number of tests.

Among the three approaches, we believe restricting the sample by comprehensive contact tracing would work best in minimizing measurement error in testing. Further, since testing strategies have changed over time for some countries, limiting the sample to days when contact tracing was done as a consistent testing strategy, will also be a good robustness check for our results. First, we conduct our analysis using the full sample. Then, we check the robustness of the findings using the restricted sample of countries (and days) that conducted comprehensive contact tracing as a consistent testing strategy.

One other data concern is that policy stringency measures from OxCGRT capture only the restrictions imposed, and not how they are enforced or the behavior of the citizens. Our estimates, therefore, are net of enforcement, compliance, and mitigating self-disciplining behavior of the citizens. One of the nine indicators used to construct the stringency index is a measure of the coordinated presence of public awareness campaigns about the virus. Any change in the citizens' self-disciplining due to public awareness campaigns should be captured by this component of the stringency measure. Therefore, the stringency effect not only reflects citizens'

response to policy measures put in place but also any self-disciplining effect. The available data do not permit us to disentangle the two.

## 2.3 Empirical strategy

Investigating the causal impact of the level of stringency on the mobility indicators and COVID-19 outbreak growth rate variables present a few empirical challenges. While we do not provide a theoretical model to mathematically breakdown the causal mechanism, Keppo et al. (2020) [41] extend the epidemiological SIR model to a "behavioral SIR model" and is a good resource for anyone looking for a theoretical construct. First, governments around the world enacted these measures in response to the disease situation in their countries. Therefore, ordinary least squares estimation (OLS) of the associations between the stringency of the policy measures and the outbreak growth rates could be driven by reverse causality—countries with worse disease situations had to enact more stringent measures to control the contagion. Similarly, even without the announced restrictions, countries with a higher proportion of circumspect population might see a decrease in both mobility and disease spread. The governments in these countries might have responded to the expectation this could have placed on the government to support their citizens. Country or time fixed effects will not be able to account for the changes in expectations people have from their government or actions of the government in response to these expectations across time. There is also considerable variation in how well and how soon governments might have reacted to the disease environment in the country. That is, the extent to reverse causality also varies by country. Further, as discussed in section 2.2, the outcome measures likely suffer from non-classical measurement errors. For example, less educated countries might be less stringent and might also have larger measurement errors in recording cases and deaths. All these factors will bias the OLS estimates.

To address these concerns, we opt for an instrumental variable (IV) approach. We use the level of stringency in countries other than country $c$ on date $t$, to predict the level of stringency of the restrictions in the country $c$ on date $t$. The rationale is that governments, in deciding the level of stringency of the restriction, looked not only at the disease condition in their own country but also what they expected would happen if they did not impose stricter measures. Since there was no way for them to predict the counterfactual scenario, they looked at the situation of other countries. In particular, they observed the actions other countries in the world were taking. If a country observed that other countries around the world were imposing strict restrictions, it could have been also inclined to enact stricter restrictions regardless of the disease situation at home. Since a country did not observe the private signal of other countries about how bad they expected the situation to become, the country used the observable decision of other countries to inform its own decision. This is at least likely to be true in the initial stages of the COVID-19 pandemic (see [19]), the timeframe in concern for this study. However, lockdowns and other such extreme restrictions are not sustainable for long, especially in developing countries, with countries soon facing pressure to gradually open up while balancing various health and socio-economic concerns [42]. So, the level of stringency of the policy measures in a country $c$ at time $t$ must be correlated with the stringency of the policy measures in the rest of the world, satisfying the relevance requirement for the IV.

While the day-to-day variations in the extent of governments-imposed restrictions in the rest of the world might influence a country's propensity to impose mobility and activity restrictions, it should not, at least in the short-run, significantly affect the level of activity and the growth rate of confirmed cases or deaths in the country. However, one possibility is if in the presence of a lead time prior to implementing travel bans, asymptomatic individuals traveled to avoid anticipated restrictions in the home country, and thereby seeded an outbreak in

another country, thus directly driving cases/deaths. While this possibility is indeed difficult to negate using available data, we try to minimize this possibility by comparing several different geographical definitions of our instrument. We use the World Bank's classification of world regions and sub-regions for this exercise. In the preferred specification, we construct the instrument measuring the average level of stringency at time $t$ in countries in our sample excluding all countries in the same sub-region $s$ as country $c$ to minimize the effects of spill-overs of infections across borders. Other definitions we explore are (1) World stringency minus country stringency, (2) Region stringency minus country stringency, (3) Sub-region stringency minus country stringency, and (4) World stringency minus region stringency. The results using these alternative instruments are reported in the S1 Appendix.

Another scenario where the instrument will not be valid is if individual behavior is domi-nantly affected not only by the news she receives in her home country but also by the news she receives from around the world. That is if her behavior is affected directly by stringency poli-cies in other countries. However, as the event study presented in Fig 1 below shows, this has not been the case.

In Fig 1, $Day$ = 0 represents the day of national lockdown while the red line to the left of $Day$ = 0 represents the average number of days prior when the date of national lockdown was announced/recommended. As can be seen, changes in mobility are in a sharp response to changing stringency in the home country. The change in mobility is most stark after the date of national lockdown was announced/recommended, remaining fairly constant prior to that, and reaching its peak almost concurrently with the day of national lockdown. However, note that given the differential timing of when countries implemented actions to restrict activity, $Day$ = 0 varies across countries given there was not a singular date when countries imple-mented these measures. So one can argue for the case that the change in mobility is picking up both the response to the home country's action as well as other countries' activity restricting policies prior to the home country's implementation. In order to check for this possibility, we plot separate event study graphs for each country and see that the change in mobility is most stark after the date of lockdown was announced/recommended at home country, and is not driven by lockdowns of other countries. This suggests that individual behavior was dominantly affected by the news she receives in her home country. Therefore, the exclusion restriction is likely to hold.

The first stage of our preferred 2SLS specification is as follows:

$$\text{First Stage}: Stringency_{c,t} = \alpha1 + \beta1 \times Stringency_{w-s,t} + \theta1_c + \delta1_{t-i} + \varepsilon1_{c,t} \tag{1}$$

where, $Stringency_{c,t}$ is the level of stringency of the measures at time $t$ in country $c$. $Stringency_{w-s,t}$ measures the average level of stringency at time $t$ in countries in our sample excluding all coun-tries in the sub-region $s$ of country $c$. We also tried instruments by ranking each country's GDP/capita and categorizing them into 10 quantile groups. Then two instruments were con-structed as follows: (i) Quantile group minus country average, and (ii) Other quantile groups minus country quantile group average. We get similar results using these instruments as our preferred specification and are available upon request. Excluding the sub-region also provides more overall variation to the instrument, even though under this specification it does not vary within a sub-region (at time $t$). Note that even with the other definitions of the instrument, where it varies within sub-regions, our results remain consistent in significance and direction, as can be seen in the S1 Appendix. $\theta1_c$ controls for time-invariant unobservables and differ-ences across countries that capture factors like differential measurement errors in outcomes variables, levels of health and health infrastructure, times at which the first case was detected in different countries, and so on. $\delta1_{t-i}$ controls for effects that are associated with days since the

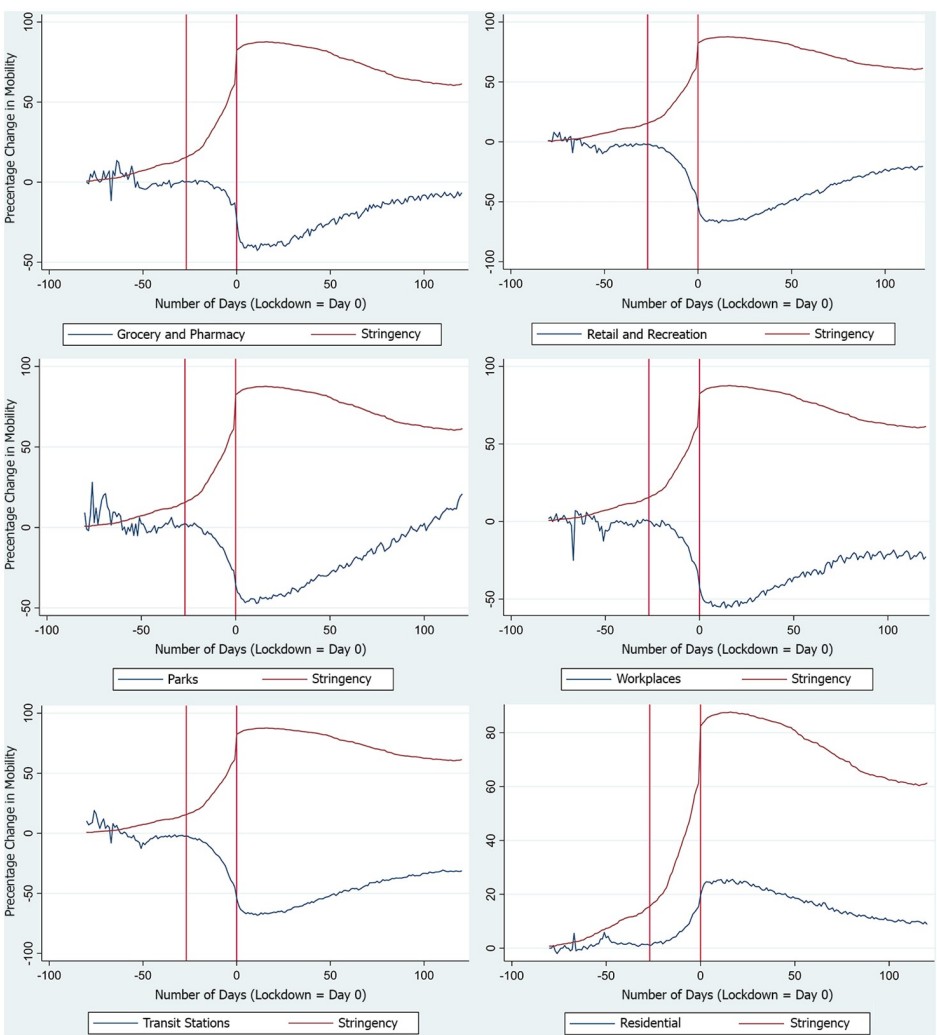

**Fig 1. Event study of days since country lockdown(= 0) and mobility measures.** Event study of 107 countries with national lockdown. Percentage change in mobility is in reference to the median value of mobility for the corresponding day of the week during the 5 weeks of January 3, 2020, to February 6, 2020. *Day* = 0 represents the day of national lockdown in the home country. The red line to the left of *Day* = 0 represents the average number of days prior to lockdown when the date of national lockdown was announced/recommended.

first confirmed case in the country, where *i* is the day of the first confirmed case. We believe $\delta 1_{t-i}$ does a better job at capturing the time-varying unobservable factors that might affect stringency across countries. This is because how the disease spreads within a country depends on when the first confirmed case was detected. For example, since the first confirmed case in China was much earlier than in the United States of America, there is no reason why both countries will have a similar level of unobservable factors affecting *Stringency$_{c,t}$* on February 15, 2020. Both the fixed effects, $\theta 1_c$ and $\delta 1_{t-i}$, thus attempts to control for any correlation of unobservables with other countries' stringency, and own country cases/deaths/mobility.

We then use the predicted values of *Stringency$_{c,t}$* in:

$$\text{Second Stage}: Y_{c,t} = \alpha 2 + \beta 2 \times \widehat{Stringency}_{c,t} + \theta 2_c + \delta 2_{t-i} + \varepsilon 2_{c,t} \qquad (2)$$

where $Y_{c,t}$ is any of the mobility or outbreak growth rate outcomes for country *c* at time *t*. In

some of our second stage specifications, we replace $\widehat{Stringency}_{c,t}$ with $\widehat{Stringency}_{c,t-7}$, $\widehat{Stringency}_{c,t-14}$, $\widehat{Stringency}_{c,t-21}$, or $\widehat{Stringency}_{c,t-28}$ to account for the possibility that the impact of a change in stringency on the number of confirmed cases and deaths might show up after a lag. We cluster the standards errors at the level of the country. Note that our instrument does not correct for non-classical measurement errors. However, in the preferred specification, excluding all other countries in the sub-region can minimize the chances of the measurement error in the instrument being correlated with measurement error in the endogenous variable. In the case of skewed or one-sided measurement error in the dependent variable, as is likely our case with the under-reporting of cases and deaths, Millimet and Parmeter (2019) [32] proposes using stochastic frontier analysis (SFA) or nonlinear least squares (NLLS) estimation to correct for such bias. Similar approaches has been used by Hofler and List (2004) [43] and Kumbhakar et al. (2012) [44] to correct for systematic over- or under-bidding in auctions, and by Anthopolos and Becker (2010) [45] to correct for undercounting in infant mortality data. As a robustness check, we repeat our analyses using SFA and NLLS estimators and find our results to be qualitatively similar to the ones reported using 2SLS. SFA and NLLS results are available upon request.

A decrease in mobility and activity, due to the stricter restrictions, may not necessarily decrease the growth of deaths due to the virus. For example, if those infected transfer it to others in and around their living quarters, infections and deaths may not decrease even if mobility does. To understand better the effectiveness of decreasing mobility and activity on the contagion, we estimate a three-stage recursive conditional mixed-process (CMP) model from Roodman (2011) [46]. The process is akin to a 3-Stages Least Square methodology, and similar to the 2SLS estimator assumes $Stringency_{w-s,t}$ to be exogenous. To understand the intuition behind the process, note that,

$$\frac{d\ (growth\ rate)}{d\ (mobility)} = \frac{d\ (growth\ rate)}{d\ (stringency\ index)} \Big/ \frac{d\ (mobility)}{d\ (stringency\ index)}$$

That is, the ratio of the causal IV estimate of the impact of the stringency index on the growth rates to the impact of the stringency index on mobility is an estimate of how mobility affected the growth rates of cases or deaths in different countries. The system of equations is as follows:

$$Stringency_{c,t} = a1 + b1 \times Stringency_{w-s,t} + \gamma1_c + \tau1_{t-i} + \epsilon1_{c,t} \tag{3}$$

$$Mobility_{c,t} = a2 + b2 \times \widehat{Stringency}_{c,t} + \gamma2_c + \tau2_{t-i} + \epsilon2_{c,t} \tag{4}$$

$$Growth\ Rate_{c,t} = a3 + b3 \times \widehat{Mobility}_{c,t} + \gamma3_c + \tau3_{t-i} + \epsilon3_{c,t} \tag{5}$$

This allows us to compare how changes in mobility, due to stringent policy measures, affected the growth rates of cases to tests or deaths to tests ratios in different countries. We use mobility at public transport transit stations for this analysis.

## 3 Results

The mobility and activity restrictions enacted by countries around the world aimed at containing the contagion by limiting human-to-human contact. However, it is not obvious whether these restrictions actually limited mobility and activity; it depended on people's will and ability to observe these restrictions and their government's ability to enforce them. For example, multiple factors including, but not limited to, the level of education, trust in the government, and ability to maintain basic consumption expenditure without working, affect the extent to which

citizens of a country might observe the restrictions. In Table 2, we begin by examining the impact of these restrictions on mobility. The dependent variables in columns (1) to (6) are the percentage changes in mobility in areas of the country as compared to the median value for the corresponding day of the week, during the 5 weeks of January 3, 2020, to February 6, 2020. The first five panels of the table present the association between these dependent variables and the stringency of the restrictions in the country at distinct points in time.

The estimated coefficient for Stringency Index (Lag 0) reports the association between mobility and contemporaneous restrictions. Similarly, coefficients for Stringency Index (Lag 7), Stringency Index (Lag 14), Stringency Index (Lag 21), and Stringency Index (Lag 28) report the association of the mobility measures with the stringency of the restrictions seven, fourteen, twenty-one, and twenty-eight days ago, respectively. All specifications include country fixed-effects and the number of days since the first case fixed effect, and we cluster the standard errors at the country level.

Two observations stand out. First, the restrictions had the intended impact—countries with stricter restrictions observed a higher reduction in mobility in public areas and an increase in time spent in residential areas. This is consistent with the associations between restrictions and mobility that [12, 14, 16, 17] report. Second, as expected, contemporaneous restrictions matter more than past restrictions. The magnitude of the association of mobility measures falls with increasing lagged days of stringency of the restrictions.

The next five panels of the table present the results from the instrumental variable (IV) approach. As we discuss in Section 2, we use the level of stringency of the restrictions in countries in the rest of the world to predict the level of stringency in a country. The rationale, once again, is that countries, while deciding on the level of stringency responded not only to the disease situation at the home country but also to how it was expected to evolve. To predict how the situation would have evolved and what the optimal level of stringency might have been, every country looked at the rest of the countries in the world. Therefore, while the level of stringency in the rest of the world affected the stringency of the restrictions in a country, it did not affect the mobility and the disease situation in the country directly. That is, the exclusion restriction is likely to be satisfied. We use several definitions of the instrumental variable, all of which yield similar results. We present the results from using alternative instruments in S5 and S6 Tables in S1 Appendix. In what follows, we present results from our preferred IV specification where we use the stringency in the countries outside the sub-region to which the country belongs. Excluding countries from the sub-regions minimize the chances of the stringency in other countries affecting the mobility or spread of the disease in the country through pathways other than affecting the country's restriction stringency. The first stage F-stats are provided under each estimation and are well above the conventionally accepted threshold of 10 (for the case of a single endogenous regressor; see, [47]), indicating that the instrument is relevant and not weak. Compared to the association results in the first five panels, the IV causal estimates are slightly larger in magnitude. But the two broad observations remain unchanged —countries with stricter restrictions observed higher reduction in mobility, and contemporaneous restrictions matter more than past restrictions.

Next, in Table 3, we examine the impact of the level of stringency of the restrictions on the 7-day moving average growth rates of the numbers of tests conducted, confirmed cases, cases to tests ratio, deaths attributed to COVID-19, and deaths to tests ratio across time in different countries. The first five panels present the associations for comparison, but the discussion hereon will focus on the IV results. Compared to Table 2 where the contemporaneous restrictions had the largest impact on mobility, the stringency of the measures seven days and fourteen days ago have a larger impact on the growth rate of confirmed cases and deaths attributed to COVID-19. Given the current scientific understanding that the virus has an incubation and

**Table 2. Impact of stricter restrictions on mobility.**

| VARIABLES | (1) Retail Recreation | (2) Grocery Pharmacy | (3) Parks | (4) Transit Stations | (5) Workplaces | (6) Residential |
|---|---|---|---|---|---|---|
| | | | OLS | | | |
| Stringency Index (Lag 0) | -0.89*** | -0.53*** | -0.73*** | -0.81*** | -0.68*** | 0.31*** |
| | (0.03) | (0.03) | (0.08) | (0.03) | (0.03) | (0.01) |
| Mean of DV | -33.35 | -15.70 | -7.38 | -38.39 | -27.22 | 13.14 |
| Stringency Index (Lag 7) | -0.75*** | -0.48*** | -0.64*** | -0.68*** | -0.56*** | 0.26*** |
| | (0.03) | (0.03) | (0.08) | (0.03) | (0.03) | (0.01) |
| Mean of DV | -33.70 | -15.25 | -7.54 | -38.79 | -27.50 | 13.28 |
| Stringency Index (Lag 14) | -0.55*** | -0.35*** | -0.47*** | -0.48*** | -0.39*** | 0.19*** |
| | (0.03) | (0.02) | (0.07) | (0.03) | (0.02) | (0.01) |
| Mean of DV | -34.18 | -15.50 | -7.73 | -39.31 | -27.88 | 13.45 |
| Stringency Index (Lag 21) | -0.35*** | -0.21*** | -0.31*** | -0.29*** | -0.22*** | 0.11*** |
| | (0.03) | (0.02) | (0.07) | (0.03) | (0.02) | (0.01) |
| Mean of DV | -35.05 | -16.00 | -8.03 | -40.28 | -28.65 | 13.77 |
| Stringency Index (Lag 28) | -0.17*** | -0.08*** | -0.18*** | -0.13*** | -0.05** | 0.05*** |
| | (0.03) | (0.02) | (0.07) | (0.02) | (0.02) | (0.01) |
| Mean of DV | -36.16 | -16.77 | -8.41 | -41.52 | -29.65 | 14.19 |
| | | | 2SLS: Excluding Subregion IV | | | |
| Stringency Index (Lag 0) | -1.12*** | -0.69*** | -0.60*** | -1.05*** | -0.92*** | 0.38*** |
| | (0.05) | (0.04) | (0.09) | (0.04) | (0.04) | (0.02) |
| Mean of DV | -33.34 | -15.07 | -7.38 | -38.39 | -27.22 | 13.14 |
| F-Stat | 268.00 | 267.95 | 268.00 | 268.00 | 268.00 | 267.67 |
| Stringency Index (Lag 7) | -0.93*** | -0.63*** | -0.47*** | -0.88*** | -0.79*** | 0.33*** |
| | (0.04) | (0.04) | (0.09) | (0.04) | (0.04) | (0.02) |
| Mean of DV | -33.70 | -15.24 | -7.53 | -38.78 | -27.50 | 13.27 |
| F-Stat | 286.24 | 286.14 | 286.24 | 286.24 | 286.24 | 285.40 |
| Stringency Index (Lag 14) | -0.66*** | -0.47*** | -0.27*** | -0.62*** | -0.57*** | 0.24*** |
| | (0.04) | (0.03) | (0.09) | (0.04) | (0.04) | (0.02) |
| Mean of DV | -34.17 | -15.50 | -7.73 | -39.31 | -27.88 | 13.45 |
| F-Stat | 294.44 | 294.30 | 294.45 | 294.44 | 294.44 | 293.05 |
| Stringency Index (Lag 21) | -0.39*** | -0.28*** | -0.09 | -0.35*** | -0.31*** | 0.13*** |
| | (0.05) | (0.03) | (0.09) | (0.04) | (0.04) | (0.02) |
| Mean of DV | -35.05 | -15.99 | -8.03 | -40.28 | -28.65 | 13.77 |
| F-Stat | 303.31 | 303.16 | 303.31 | 303.31 | 303.31 | 300.92 |
| Stringency Index (Lag 28) | -0.14*** | -0.10*** | 0.05 | -0.11*** | -0.07* | 0.04** |
| | (0.05) | (0.03) | (0.08) | (0.04) | (0.04) | (0.02) |
| Mean of DV | -36.16 | -16.76 | -8.40 | -41.52 | -29.65 | 14.19 |
| F-Stat | 312.68 | 312.58 | 312.67 | 312.68 | 312.68 | 311.26 |
| Observations (Lag 0) | 18,322 | 18,317 | 18,321 | 18,322 | 18,322 | 18,246 |
| Observations (Lag 7) | 18,119 | 18,114 | 18,118 | 18,119 | 18,119 | 18,043 |
| Observations (Lag 14) | 17,850 | 17,845 | 17,849 | 17,850 | 17,850 | 17,774 |
| Observations (Lag 21) | 17,398 | 17,393 | 17,397 | 17,398 | 17,398 | 17,322 |
| Observations (Lag 28) | 16,792 | 16,787 | 16,791 | 16,792 | 16,792 | 16,716 |
| Number of country | 127 | 127 | 127 | 127 | 127 | 127 |
| Fixed Effects | | | Country; Days since first case | | | |

Robust standard errors clustered at the country level.

* $p < 0.10$;

** $p < 0.05$;

*** $p < 0.01$.

**Table 3. Impact of stricter restrictions on 7-day moving average growth rates.**

| VARIABLES | (1) Tests | (2) Cases | (3) Cases:Tests | (4) Deaths | (5) Deaths:Tests |
|---|---|---|---|---|---|
| | | | OLS | | |
| Stringency Index (Lag 0) | -0.0000 | -0.0006*** | -0.0005* | -0.0019*** | -0.0007 |
| | (0.0001) | (0.0002) | (0.0002) | (0.0005) | (0.0005) |
| Mean of DV | 0.047 | 0.071 | 0.006 | 0.070 | 0.018 |
| Stringency Index (Lag 7) | -0.0003* | -0.0014*** | -0.0006*** | -0.0024*** | -0.0011*** |
| | (0.0002) | (0.0002) | (0.0002) | (0.0004) | (0.0004) |
| Mean of DV | 0.046 | 0.071 | 0.006 | 0.069 | 0.017 |
| Stringency Index (Lag 14) | -0.0004*** | -0.0014*** | -0.0006*** | -0.0026*** | -0.0013*** |
| | (0.0001) | (0.0001) | (0.0001) | (0.0003) | (0.0003) |
| Mean of DV | 0.046 | 0.069 | 0.006 | 0.069 | 0.017 |
| Stringency Index (Lag 21) | -0.0004*** | -0.0012*** | -0.0005*** | -0.0022*** | -0.0012*** |
| | (0.0001) | (0.0001) | (0.0001) | (0.0002) | (0.0003) |
| Mean of DV | 0.044 | 0.063 | 0.005 | 0.067 | 0.017 |
| Stringency Index (Lag 28) | -0.0004*** | -0.0009*** | -0.0004*** | -0.0016*** | -0.0009*** |
| | (0.0001) | (0.0001) | (0.0001) | (0.0002) | (0.0002) |
| Mean of DV | 0.041 | 0.055 | 0.002 | 0.063 | 0.016 |
| | | | 2SLS: Excluding Subregion IV | | |
| Stringency Index (Lag 0) | -0.0007** | -0.0015*** | -0.0011** | -0.0060*** | -0.0050*** |
| | (0.0003) | (0.0003) | (0.0005) | (0.0013) | (0.0019) |
| Mean of DV | 0.047 | 0.071 | 0.006 | 0.070 | 0.018 |
| F-Stat | 41.899 | 267.999 | 41.899 | 65.795 | 31.513 |
| Stringency Index (Lag 7) | -0.0009*** | -0.0033*** | -0.0019*** | -0.0053*** | -0.0039*** |
| | (0.0003) | (0.0003) | (0.0003) | (0.0007) | (0.0010) |
| Mean of DV | 0.046 | 0.071 | 0.006 | 0.069 | 0.017 |
| F-Stat | 58.113 | 286.243 | 58.113 | 123.500 | 37.763 |
| Stringency Index (Lag 14) | -0.0009*** | -0.0033*** | -0.0018*** | -0.0046*** | -0.0032*** |
| | (0.0002) | (0.0002) | (0.0002) | (0.0005) | (0.0006) |
| Mean of DV | 0.046 | 0.069 | 0.006 | 0.069 | 0.017 |
| F-Stat | 78.483 | 294.437 | 78.483 | 187.783 | 55.454 |
| Stringency Index (Lag 21) | -0.0007*** | -0.0018*** | -0.0009*** | -0.0037*** | -0.0024*** |
| | (0.0002) | (0.0002) | (0.0002) | (0.0003) | (0.0004) |
| Mean of DV | 0.044 | 0.063 | 0.005 | 0.067 | 0.017 |
| F-Stat | 96.858 | 303.310 | 96.858 | 234.248 | 73.762 |
| Stringency Index (Lag 28) | -0.0006*** | -0.0013*** | -0.0006*** | -0.0026*** | -0.0018*** |
| | (0.0001) | (0.0001) | (0.0001) | (0.0002) | (0.0002) |
| Mean of DV | 0.041 | 0.055 | 0.002 | 0.063 | 0.016 |
| F-Stat | 116.105 | 312.677 | 116.105 | 261.507 | 94.749 |
| Observations (Lag 0) | 10,660 | 18,322 | 10,660 | 15,268 | 9,675 |
| Observations (Lag 7) | 10,607 | 18,119 | 10,607 | 15,215 | 9,651 |
| Observations (Lag 14) | 10,537 | 17,850 | 10,537 | 15,149 | 9,622 |
| Observations (Lag 21) | 10,400 | 17,398 | 10,400 | 15,050 | 9,584 |
| Observations (Lag 28) | 10,133 | 16,792 | 10,133 | 14,860 | 9,499 |
| Number of country | 80 | 127 | 80 | 121 | 78 |
| Fixed Effects | | | Country; Days since first case | | |

Robust standard errors clustered at the country level.

* $p < 0.10$;

** $p < 0.05$;

*** $p < 0.01$.

infection period of up to fourteen days, this is expected. Second, even if we focus only on the effect of stringency seven or fourteen days prior, there appears to be a much smaller effect on the number of tests. There is no reason why the number of tests conducted, given the testing infrastructure of a country is controlled for by the country fixed effects, would have been largely affected by a decrease in mobility. It is possible that with reduced mobility, events like accidents that require medical attention decreases reducing the pressure on the health infrastructure that could then be devoted to COVID-19 testing. However, that would have lead to an increase in testing, which is not what we observe.

However, the more stringent the measures, the lower the growth in the number of confirmed cases and deaths attributed to COVID-19. The impact on cases and deaths suggests that stricter restrictions achieved their goal of containing the contagion. The impacts on the growth rate of the two ratio variables are, as expected, smaller in magnitude but follow the same trends. Since we believe them to be better indicators, we report results with the growth rates of cases to tests and deaths to tests, hereon. As discussed in Section 2.2, in S7 Table in S1 Appendix we show that the results are robust to using restricted samples by testing approaches followed by different countries. The coefficients, when the sample is limited to countries and dates with full contact tracing as a consistent testing strategy, are similar to that of the full sample and follow the same trend.

But were restrictions equally effective across developing and developed countries, and adequate to contain the contagion? Countries with differing institutional capacities are likely to respond differently in not only adopting stringency measures [48] but also in their subsequent role in curbing mobility and in containing the contagion. Heterogeneity analysis by demography, the status of the health infrastructure, and governance indicators will help us understand the mechanisms and the role of other institutional and cultural factors. To find out, we split the sample of countries at the median for a range of characteristics and repeat the analysis. Another approach would be to include all these different dimensions of heterogeneity in one regression. However, setting aside the multicollinearity concerns that would arise from such an approach, we are not interested in individual heterogeneity coefficients along these dimensions but rather what they suggest collectively.

We present the heterogeneity in the impact of stringency on mobility in Tables 4 and 5, along with its coefficients plot in Fig 2 presented below. The first and last three columns in each panel of the tables report the impact of imposing stricter restrictions on mobility in countries below and above the median along the different dimensions. Comparing column (1) with column (4), column (2) with column (5), and column (3) with column (6), stricter restrictions had a larger marginal effect in limiting mobility in densely populated, less educated, poorer, more unequal, more polluted countries with younger but unhealthier populations and worse health infrastructure. From their description, and affirmed by the segregated summary statistics presented in S3 Table in S1 Appendix, these characteristics describe the relatively less-developed countries in the sample. The restrictions also worked better in more democratic countries, with better government effectiveness and lower perceived levels of corruption. Finally, it should be noted from Fig 2 that not all two-point estimates are statistically significantly different from each other; but rather the purpose of the exercise is to point towards general trends of the coefficients over the different mobility measures.

However, this stronger effect of stringency on mobility does not imply that the relatively less-developed countries contained the contagion better. First, it is important to note that upon announcement of the lock-downs, many less-developed countries witnessed large migration of urban migrant workers back to their homes in rural areas before the lockdown came into effect, relevant for the timeframe explored in this study (see, for example [49], for the case in India). With limited mobility (or mobility not captured in the Google data) in their rural

**Table 4. Heterogenous impact of stricter restrictions on mobility 1.**

| VARIABLES | (1) Transit Stations | (2) Workplaces | (3) Residential | (4) Transit Stations | (5) Workplaces | (6) Residential |
|---|---|---|---|---|---|---|
| | < Median | | | > Median | | |
| | Population Density | | | | | |
| Stringency Index (Lag 14) | -0.42*** | -0.57*** | 0.21*** | -0.63*** | -0.59*** | 0.26*** |
| | (0.06) | (0.05) | (0.02) | (0.05) | (0.05) | (0.02) |
| Observations | 8,590 | 8,590 | 8,564 | 9,262 | 9,262 | 9,212 |
| Number of country | 63 | 63 | 63 | 64 | 64 | 64 |
| Mean of DV | -37.12 | -26.09 | 12.76 | -41.35 | -29.55 | 14.09 |
| F-Stat | 231.90 | 231.90 | 231.61 | 138.16 | 138.16 | 137.49 |
| | Primary Education | | | | | |
| Stringency Index (Lag 14) | -0.48*** | -0.50*** | 0.25*** | -0.64*** | -0.56*** | 0.23*** |
| | (0.05) | (0.04) | (0.02) | (0.05) | (0.04) | (0.02) |
| Observations | 5,504 | 5,504 | 5,479 | 12,348 | 12,348 | 12,297 |
| Number of country | 39 | 39 | 39 | 88 | 88 | 88 |
| Mean of DV | -46.99 | -30.94 | 17.13 | -35.89 | -26.52 | 11.81 |
| F-Stat | 282.80 | 282.80 | 284.34 | 200.16 | 200.16 | 198.94 |
| | Poverty Head Count | | | | | |
| Stringency Index (Lag 14) | -0.55*** | -0.49*** | 0.21*** | -0.68*** | -0.62*** | 0.29*** |
| | (0.05) | (0.04) | (0.02) | (0.06) | (0.03) | (0.03) |
| Observations | 8,209 | 8,209 | 8,207 | 9,643 | 9,643 | 9,569 |
| Number of country | 56 | 56 | 56 | 71 | 71 | 71 |
| Mean of DV | -36.85 | -29.54 | 11.21 | -41.41 | -26.47 | 15.37 |
| F-Stat | 157.08 | 157.08 | 157.08 | 124.40 | 124.40 | 123.23 |
| | Gini Index | | | | | |
| Stringency Index (Lag 14) | -0.53*** | -0.51*** | 0.22*** | -0.66*** | -0.61*** | 0.27*** |
| | (0.06) | (0.03) | (0.02) | (0.05) | (0.04) | (0.03) |
| Observations | 7,851 | 7,851 | 7,849 | 10,001 | 10,001 | 9,927 |
| Number of country | 55 | 55 | 55 | 72 | 72 | 72 |
| Mean of DV | -34.65 | -26.89 | 9.92 | -42.97 | -28.66 | 16.23 |
| F-Stat | 124.98 | 124.98 | 124.98 | 175.87 | 175.87 | 174.51 |
| | Age 65 & Older | | | | | |
| Stringency Index (Lag 14) | -0.67*** | -0.56*** | 0.26*** | -0.53*** | -0.50*** | 0.24*** |
| | (0.05) | (0.05) | (0.02) | (0.05) | (0.05) | (0.02) |
| Observations | 8,623 | 8,623 | 8,572 | 9,229 | 9,229 | 9,204 |
| Number of country | 63 | 63 | 63 | 64 | 64 | 64 |
| Mean of DV | -39.75 | -25.05 | 14.47 | -38.91 | -30.53 | 12.50 |
| F-Stat | 287.80 | 170.79 | 307.09 | 135.07 | 91.81 | 108.58 |
| | PM2.5 | | | | | |
| Stringency Index (Lag 14) | -0.54*** | -0.50*** | 0.23*** | -0.62*** | -0.55*** | 0.25*** |
| | (0.06) | (0.05) | (0.03) | (0.05) | (0.05) | (0.02) |
| Observations | 8,572 | 8,572 | 8,551 | 9,280 | 9,280 | 9,225 |
| Number of country | 59 | 59 | 59 | 68 | 68 | 68 |
| Mean of DV | -39.14 | -30.17 | 12.99 | -39.48 | -25.77 | 13.88 |
| F-Stat | 194.59 | 145.15 | 153.11 | 101.77 | 32.78 | 79.06 |
| Fixed Effects | Country; Days since first case | | | | | |

Robust standard errors clustered at the country level.

* $p < 0.10$;

** $p < 0.05$;

*** $p < 0.01$.

**Table 5. Heterogenous impact of stricter restrictions on mobility 2.**

| VARIABLES | (1) Transit Stations | (2) Workplaces | (3) Residential | (4) Transit Stations | (5) Workplaces | (6) Residential |
|---|---|---|---|---|---|---|
| | < Median | | | > Median | | |
| | Corruption Perception Index | | | | | |
| Stringency Index (Lag 14) | -0.54*** | -0.45*** | 0.20*** | -0.70*** | -0.66*** | 0.28*** |
| | (0.05) | (0.05) | (0.02) | (0.05) | (0.05) | (0.02) |
| Observations | 7,792 | 7,792 | 7,773 | 10,060 | 10,060 | 10,003 |
| Number of country | 57 | 57 | 57 | 70 | 70 | 70 |
| Mean of DV | -38.50 | -25.89 | 14.14 | -39.94 | -29.43 | 12.91 |
| F-Stat | 210.24 | 210.24 | 209.84 | 160.18 | 160.18 | 159.78 |
| | Democracy Score | | | | | |
| Stringency Index (Lag 14) | -0.52*** | -0.55*** | 0.25*** | -0.66*** | -0.61*** | 0.25*** |
| | (0.06) | (0.06) | (0.03) | (0.05) | (0.05) | (0.02) |
| Observations | 8,077 | 8,077 | 8,058 | 9,775 | 9,775 | 9,718 |
| Number of country | 59 | 59 | 59 | 68 | 68 | 68 |
| Mean of DV | -37.78 | -24.46 | 13.29 | -40.58 | -30.71 | 13.58 |
| F-Stat | 186.80 | 186.80 | 186.41 | 147.05 | 147.05 | 146.10 |
| | Government Effectiveness | | | | | |
| Stringency Index (Lag 14) | -0.57*** | -0.48*** | 0.21*** | -0.68*** | -0.64*** | 0.27*** |
| | (0.05) | (0.05) | (0.02) | (0.05) | (0.05) | (0.02) |
| Observations | 8,810 | 8,810 | 8,784 | 9,042 | 9,042 | 8,992 |
| Number of country | 64 | 64 | 64 | 63 | 63 | 63 |
| Mean of DV | -38.30 | -26.18 | 13.94 | -40.30 | -29.54 | 12.97 |
| F-Stat | 185.93 | 185.93 | 185.54 | 150.79 | 150.79 | 150.31 |
| | Hospital Beds per 100k | | | | | |
| Stringency Index (Lag 14) | -0.61*** | -0.53*** | 0.23*** | -0.66*** | -0.62*** | 0.26*** |
| | (0.05) | (0.05) | (0.02) | (0.05) | (0.05) | (0.03) |
| Observations | 8,048 | 8,048 | 8,016 | 9,804 | 9,804 | 9,760 |
| Number of country | 58 | 58 | 58 | 69 | 69 | 69 |
| Mean of DV | -42.92 | -28.25 | 16.01 | -36.35 | -27.58 | 11.34 |
| F-Stat | 335.75 | 335.75 | 336.19 | 143.69 | 143.69 | 142.33 |
| | Handwashing Facilities | | | | | |
| Stringency Index (Lag 14) | -0.63*** | -0.49*** | 0.23*** | -0.64*** | -0.60*** | 0.25*** |
| | (0.06) | (0.07) | (0.02) | (0.05) | (0.04) | (0.02) |
| Observations | 3,891 | 3,891 | 3,891 | 13,961 | 13,961 | 13,885 |
| Number of country | 29 | 29 | 29 | 98 | 98 | 98 |
| Mean of DV | -34.38 | -20.86 | 13.65 | -40.69 | -29.84 | 13.39 |
| F-Stat | 202.92 | 202.92 | 202.92 | 222.25 | 222.25 | 220.55 |
| | Cardiovascular Diseases Death Rate | | | | | |
| Stringency Index (Lag 14) | -0.51*** | -0.52*** | 0.25*** | -0.62*** | -0.53*** | 0.22*** |
| | (0.06) | (0.06) | (0.03) | (0.05) | (0.05) | (0.02) |
| Observations | 8,944 | 8,944 | 8,906 | 8,908 | 8,908 | 8,870 |
| Number of country | 62 | 62 | 62 | 65 | 65 | 65 |
| Mean of DV | -43.62 | -32.11 | 15.46 | -34.99 | -23.63 | 11.43 |
| F-Stat | 124.63 | 124.63 | 123.19 | 201.92 | 201.92 | 200.94 |
| Fixed Effects | Country; Days since first case | | | | | |

Robust standard errors clustered at the country level.

* $p < 0.10$;

** $p < 0.05$;

*** $p < 0.01$.

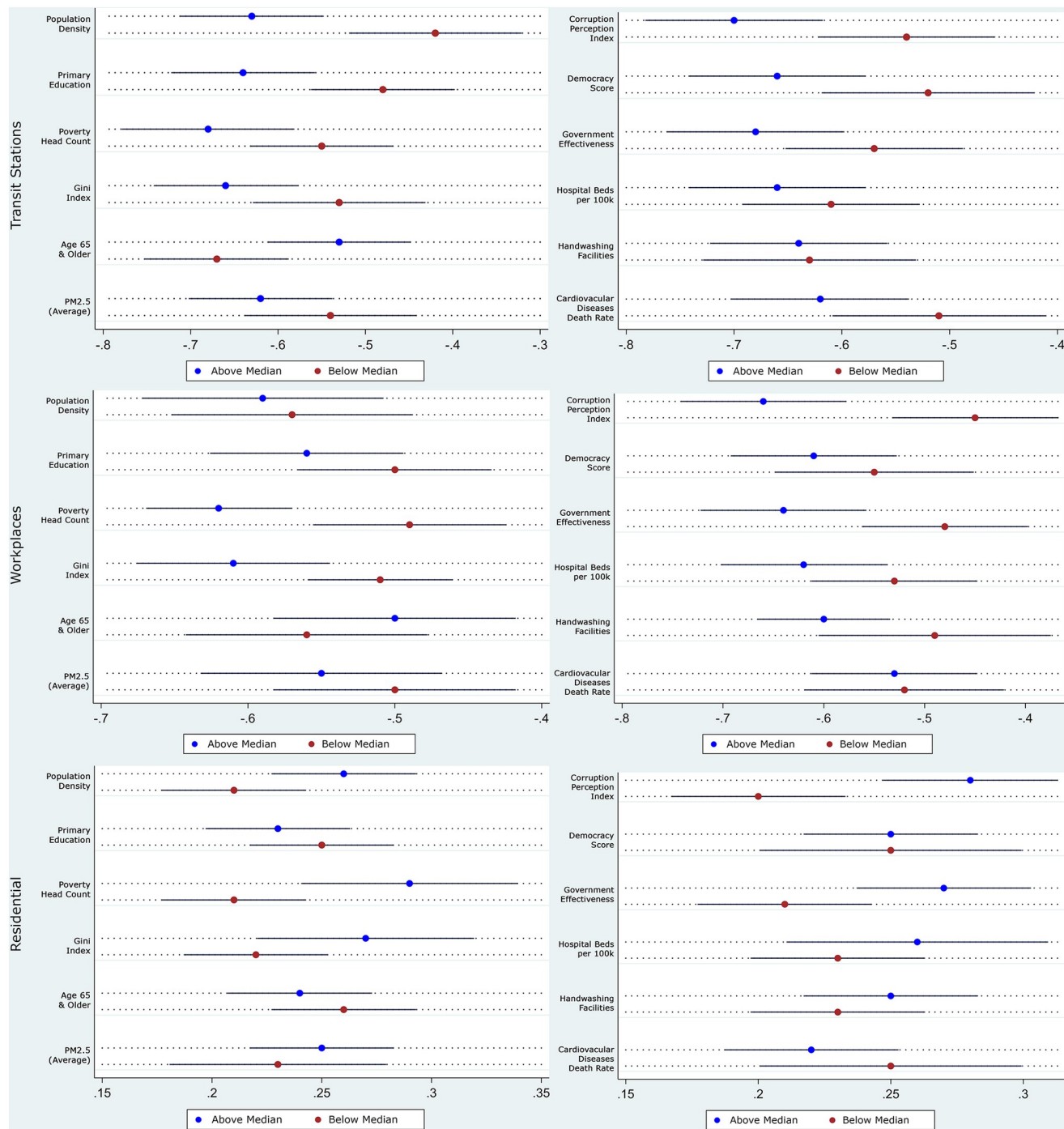

**Fig 2. Coefficient plots of hetegeneous impact of stricter restriction on mobility.** Coefficient plots are constructed using results from Tables 4 and 5 with 90% confidence intervals.

homes, this can contribute to the stronger stringency effect on mobility for less-developed countries, while not translating to better contagion containment. Second, it may be that people in more developed countries were already socially distancing even in the absence of these restrictions [50]. From S3 Table in S1 Appendix we know that compliance to mobility

restrictions was potentially lower in less-developed countries. This is supported by several other studies. For example, Ali et al. (2020) [51] report that compliance to lockdown measures for most people in Bangladesh was conditional on proper relief distribution by the government (the lack of which, due to weak institutional capacity, lead to the effective end of lockdown). Choudhury et al. (2020) [52] also show that food security policies played a crucial role in ensuring lockdown compliance in India. Thus, a lower self-disciplining behavior of the citizens in less-developed countries may be leading to the larger marginal effects of stringency on mobility in the countries. Think of this as being similar to an early point on a diminishing marginal returns curve, resulting in a larger marginal effect, but not with an overall greater reduction in mobility. Similarly, it is also possible that countries with a population in better health and adequate health infrastructure, handled the infections better, even if the restrictions were not stringent or if the populations were lax about observing them (examples include Sweden, Norway, and Germany).

We see this in Tables 6 and 7, and its coefficients plot in Fig 3 presented below. As opposed to the results in Tables 4 and 5, stricter measures contained the contagion better in richer, more educated, more equal, less-polluted countries with older but healthier populations and better health infrastructure. From the description, the countries appear to be the more developed countries in the sample. These results are partly in contrast with association results from [14] that finds the correlation between stricter pandemic policies and lower future mortality growth was more pronounced in countries with a greater proportion of the elderly population, higher density, greater proportion of employees in vulnerable occupations, greater democratic freedom, more international travels, and further distance from the equator. The differences in our findings highlight the need to distinguish causal effects of these restrictions from associations. Not surprisingly, the restrictions also worked better in more democratic countries, with better government effectiveness and lower perceived levels of corruption. The results from Tables 4–7 taken together suggest that even though stricter restrictions worked better at limiting mobility in relatively less developed countries, it did not translate into better control of the contagion. Once again, restricting the sample to contain only countries and days with full contact tracing as a consistent testing strategy does not change the results. The results are available upon request.

As explained in Section 2, a decrease in mobility and activity, due to the stricter restrictions, does not necessarily reduce the growth of the deaths due to the virus. To understand the effectiveness of decreasing mobility and activity on the contagion, we next report our results from the three-stage recursive conditional mixed-process (CMP) model described in Eqs (3) to (5). We use mobility at public transport transit stations for the analysis. Using alternative measures of mobility, except for mobility around parks, produces similar results. The results are presented in Table 8, with its coefficients plot in Fig 4 presented below. From the table, comparing the coefficients in column (1) with column (3), column (2) with column (4), column (5) with column (7), and column (6) with column (8), it is clear that the decrease in mobility had a larger effect in more developed countries that are more equal, have less poverty, are more educated, less polluted with better health infrastructure and governance. For example, a unit decrease in mobility in countries with more than the median score for the Corruption Perception Index (cleaner countries) causes a 0.0014 unit decrease in the growth of confirmed cases to tests ratio. The corresponding figure for countries with less than the median score for the Corruption Perception Index is insignificant and close to zero. With relatively few exceptions, the results suggest that developed countries benefited more from a reduction in mobility, in containing the growth rate of cases to tests ratio, than developing countries. This result is consistent with Barnett-Howell and Mobarak (2020) [53] who also report much lower estimated benefits of social distancing and social suppression in low-income countries. However, as

**Table 6. Heterogenous impact of stricter restrictions on growth rates 1.**

| VARIABLES | (1) Cases:Tests | (2) Deaths:Tests | (3) Cases:Tests | (4) Deaths:Tests |
|---|---|---|---|---|
| | < Median | | > Median | |
| | Population Density | | | |
| Stringency Index (Lag 14) | -0.0014*** | -0.0034*** | -0.0010*** | -0.0031*** |
| | (0.0002) | (0.0011) | (0.0003) | (0.0007) |
| Observations | 4,645 | 4,210 | 5,894 | 5,415 |
| Number of country | 36 | 35 | 44 | 43 |
| Mean of DV | 0.010 | 0.019 | 0.003 | 0.016 |
| F-Stat | 97.807 | 127.964 | 41.231 | 30.538 |
| | Primary Education | | | |
| Stringency Index (Lag 14) | -0.0005 | -0.0012*** | -0.0012*** | -0.0034*** |
| | (0.0005) | (0.0004) | (0.0002) | (0.0007) |
| Observations | 2,995 | 2,734 | 7,544 | 6,891 |
| Number of country | 24 | 24 | 56 | 54 |
| Mean of DV | 0.012 | 0.020 | 0.004 | 0.017 |
| F-Stat | 380.081 | 244.774 | 65.763 | 51.179 |
| | Poverty Head Count | | | |
| Stringency Index (Lag 14) | -0.0012*** | -0.0027*** | -0.0004 | -0.0025** |
| | (0.0002) | (0.0006) | (0.0004) | (0.0012) |
| Observations | 6,171 | 5,717 | 4,368 | 3,908 |
| Number of country | 45 | 44 | 35 | 34 |
| Mean of DV | 0.003 | 0.017 | 0.010 | 0.017 |
| F-Stat | 68.893 | 51.402 | 15.161 | 8.016 |
| | Gini Index | | | |
| Stringency Index (Lag 14) | -0.0014*** | -0.0030*** | -0.0007*** | -0.0025*** |
| | (0.0002) | (0.0007) | (0.0003) | (0.0009) |
| Observations | 5,321 | 4,825 | 5,218 | 4,800 |
| Number of country | 39 | 38 | 41 | 40 |
| Mean of DV | 0.003 | 0.020 | 0.009 | 0.015 |
| F-Stat | 48.890 | 38.987 | 29.206 | 17.008 |
| | Age 65 & Older | | | |
| Stringency Index (Lag 14) | -0.0005 | -0.0025*** | -0.0013*** | -0.0027*** |
| | (0.0003) | (0.0006) | (0.0002) | (0.0007) |
| Observations | 3,688 | 3,240 | 6,851 | 6,385 |
| Number of country | 29 | 28 | 51 | 50 |
| Mean of DV | 0.008 | 0.015 | 0.005 | 0.019 |
| F-Stat | 179.510 | 159.559 | 47.027 | 36.699 |
| | PM2.5 | | | |
| Stringency Index (Lag 14) | -0.0015*** | -0.0035*** | -0.0004 | -0.0020*** |
| | (0.0002) | (0.0008) | (0.0003) | (0.0005) |
| Observations | 6,416 | 6,110 | 4,123 | 3,515 |
| Number of country | 48 | 48 | 32 | 30 |
| Mean of DV | 0.004 | 0.018 | 0.009 | 0.017 |
| F-Stat | 59.872 | 48.921 | 23.656 | 11.577 |
| Fixed Effects | Country; Days since first case | | | |

Robust standard errors clustered at the country level.

* $p < 0.10$;

** $p < 0.05$;

*** $p < 0.01$.

**Table 7. Heterogenous impact of stricter restrictions on growth rates 2.**

| VARIABLES | (1) Cases:Tests | (2) Deaths:Tests | (3) Cases:Tests | (4) Deaths:Tests |
|---|---|---|---|---|
| | < Median | | > Median | |
| | Corruption Perception Index | | | |
| Stringency Index (Lag 14) | -0.0007 | -0.0023* | -0.0012*** | -0.0029*** |
| | (0.0005) | (0.0013) | (0.0002) | (0.0006) |
| Observations | 3,455 | 3,086 | 7,084 | 6,539 |
| Number of country | 27 | 26 | 53 | 52 |
| Mean of DV | 0.013 | 0.018 | 0.002 | 0.017 |
| F-Stat | 69.604 | 156.242 | 58.097 | 48.078 |
| | Democracy Score | | | |
| Stringency Index (Lag 14) | -0.0004 | -0.0027*** | -0.0014*** | -0.0043*** |
| | (0.0004) | (0.0008) | (0.0002) | (0.0006) |
| Observations | 3,424 | 2,825 | 7,115 | 6,800 |
| Number of country | 27 | 25 | 53 | 53 |
| Mean of DV | 0.010 | 0.019 | 0.004 | 0.017 |
| F-Stat | 73.911 | 99.776 | 49.927 | 43.538 |
| | Government Effectiveness | | | |
| Stringency Index (Lag 14) | -0.0007 | -0.0022*** | -0.0012*** | -0.0065*** |
| | (0.0006) | (0.0006) | (0.0002) | (0.0015) |
| Observations | 3,854 | 3,441 | 6,685 | 6,184 |
| Number of country | 30 | 29 | 50 | 49 |
| Mean of DV | 0.012 | 0.019 | 0.002 | 0.017 |
| F-Stat | 40.093 | 35.216 | 58.578 | 44.686 |
| | Hospital Beds per 100k | | | |
| Stringency Index (Lag 14) | -0.0006** | -0.0026*** | -0.0012*** | -0.0026*** |
| | (0.0003) | (0.0007) | (0.0002) | (0.0007) |
| Observations | 3,895 | 3,528 | 6,644 | 6,097 |
| Number of country | 30 | 29 | 50 | 49 |
| Mean of DV | 0.010 | 0.015 | 0.003 | 0.018 |
| F-Stat | 210.199 | 181.536 | 57.133 | 44.644 |
| | Handwashing Facilities | | | |
| Stringency Index (Lag 14) | -0.0001 | -0.0023** | -0.0011*** | -0.0030*** |
| | (0.0004) | (0.0010) | (0.0002) | (0.0006) |
| Observations | 1,648 | 1,429 | 8,891 | 8,196 |
| Number of country | 13 | 13 | 67 | 65 |
| Mean of DV | 0.010 | 0.018 | 0.005 | 0.017 |
| F-Stat | 369.736 | 211.754 | 66.703 | 48.252 |
| | Cardiovascular Diseases Death Rate | | | |
| Stringency Index (Lag 14) | -0.0013*** | -0.0044*** | -0.0008** | -0.0027*** |
| | (0.0002) | (0.0007) | (0.0003) | (0.0012) |
| Observations | 6,214 | 5,958 | 4,325 | 3,667 |
| Number of country | 47 | 47 | 33 | 31 |
| Mean of DV | 0.005 | 0.017 | 0.007 | 0.017 |
| F-Stat | 37.084 | 31.917 | 113.928 | 123.627 |
| Fixed Effects | Country; Days since first case | | | |

Robust standard errors clustered at the country level.

* $p < 0.10$;

** $p < 0.05$;

*** $p < 0.01$.

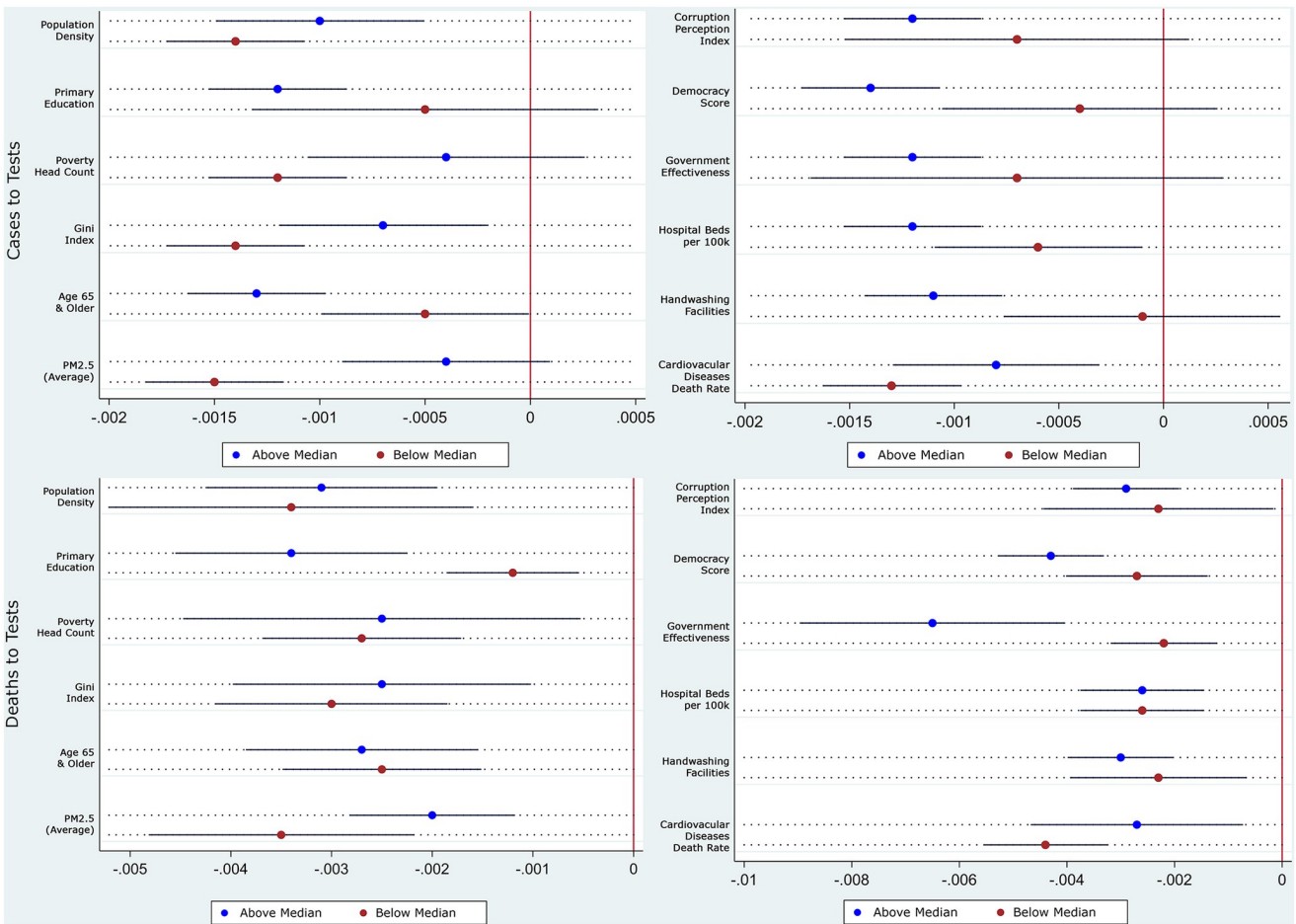

**Fig 3. Coefficient plots of hetegeneous impact of stricter restriction on growth rates.** Coefficient plots are constructed using results from Tables 6 and 7 with 90% confidence intervals.

Fig 4 visually shows, we do not find such stark differences across the heterogeneity dimensions in the case of deaths to tests ratio, where education, democracy, and government effectiveness seem to play a dominant role. The results remain consistent when restricting the sample to only countries and dates with full contact tracing as a consistent testing strategy, and are reported in S8 Table in S1 Appendix.

The heterogeneity results provide some elucidation to the possible reasons. Given that the population, on average, in relatively less-developed countries is more immunocompromised, fewer people might have been able to fight off the infections. There is a growing amount of scientific evidence that points towards people with better immune systems being able to fight SARS-CoV-2 infection better. See, for example [54]. People who can fight off the viral infection are possibly being less diagnosed, due to a shorter incubation period. Stringency measures are unable to counter immunodeficiency. This is further aggravated by the fact that stringent mobility measures lower the spread of the disease at the cost of people's economic opportunities. With higher poverty rates in developing countries, poor people may place greater value on their livelihoods relative to contracting the infection. The reduction in economic activity due to the restrictions could directly affect the daily consumption of poorer people, further compromising their immune systems. Similarly, hand-washes are not on the top of the

**Table 8. Heterogenous impact of transit station mobility on growth rates: Recursive mixed-process model.**

| VARIABLES | (1) Cases:Tests | (2) Deaths:Tests | (3) Cases:Tests | (4) Deaths:Tests | (5) Cases:Tests | (6) Deaths:Tests | (7) Cases:Tests | (8) Deaths:Tests |
|---|---|---|---|---|---|---|---|---|
| | < Median | | > Median | | < Median | | > Median | |
| | Population Density | | | | Corruption Perception Index | | | |
| Transit Stations | 0.0020*** | 0.0052*** | 0.0013 | 0.0045*** | 0.0000 | 0.0039 | 0.0014*** | 0.0041*** |
| | (0.0006) | (0.0018) | (0.0008) | (0.0014) | (0.0009) | (0.0022) | (0.0005) | (0.0012) |
| Observations | 8,619 | 8,604 | 9,356 | 9,301 | 7,820 | 7,806 | 10,155 | 10,099 |
| Mean of DV | 0.010 | 0.019 | 0.003 | 0.017 | 0.013 | 0.018 | 0.003 | 0.018 |
| | Primary Education | | | | Democracy Score | | | |
| Transit Stations | -0.0002 | 0.0019** | 0.0015** | 0.0058*** | -0.0004 | 0.0037*** | 0.0019*** | 0.0079*** |
| | (0.0013) | (0.0009) | (0.0006) | (0.0011) | (0.0007) | (0.0011) | (0.0005) | (0.0013) |
| Observations | 5,518 | 5,504 | 12,457 | 12,401 | 8,105 | 8,077 | 9,870 | 9,828 |
| Mean of DV | 0.011 | 0.020 | 0.004 | 0.017 | 0.009 | 0.019 | 0.005 | 0.017 |
| | Poverty Head Count | | | | Government Effectiveness | | | |
| Transit Stations | 0.0017** | 0.0040*** | -0.0002 | 0.0036*** | -0.0002 | 0.0033*** | 0.0015*** | 0.0081*** |
| | (0.0007) | (0.0013) | (0.0008) | (0.0014) | (0.0009) | (0.0011) | (0.0005) | (0.0015) |
| Observations | 8,290 | 8,234 | 9,685 | 9,671 | 8,087 | 8,048 | 9,888 | 9,857 |
| Mean of DV | 0.004 | 0.018 | 0.009 | 0.017 | 0.009 | 0.015 | 0.004 | 0.019 |
| | Gini Index | | | | Hospital Beds per 100k | | | |
| Transit Stations | 0.0018*** | 0.0040*** | 0.0001 | 0.0037*** | -0.0000 | 0.0039*** | 0.0020*** | 0.0040*** |
| | (0.0004) | (0.0014) | (0.0006) | (0.0012) | (0.0006) | (0.0012) | (0.0005) | (0.0014) |
| Observations | 7,942 | 7,890 | 10,033 | 10,015 | 8,087 | 8,048 | 9,888 | 9,857 |
| Mean of DV | 0.003 | 0.021 | 0.009 | 0.015 | 0.009 | 0.015 | 0.004 | 0.019 |
| | Age 65 & Older | | | | Handwashing Facilities | | | |
| Transit Stations | -0.0001 | 0.0038*** | 0.0021*** | 0.0047*** | -0.0006 | 0.0038** | 0.0014** | 0.0043*** |
| | (0.0005) | (0.0010) | (0.0005) | (0.0014) | (0.0006) | (0.0018) | (0.0006) | (0.0012) |
| Observations | 8,662 | 8,623 | 9,313 | 9,282 | 3,905 | 3,891 | 14,070 | 14,014 |
| Mean of DV | 0.007 | 0.015 | 0.006 | 0.019 | 0.009 | 0.018 | 0.006 | 0.018 |
| | PM2.5 | | | | Cardiovascular Diseases Death Rate | | | |
| Transit Stations | 0.0021*** | 0.0043*** | -0.0002 | 0.0043*** | 0.0026*** | 0.0072*** | 0.0002 | 0.0046*** |
| | (0.0006) | (0.0016) | (0.0007) | (0.0016) | (0.0008) | (0.0012) | (0.0007) | (0.0009) |
| Observations | 8,625 | 8,587 | 9,350 | 9,318 | 9,014 | 8,983 | 8,961 | 8,922 |
| Mean of DV | 0.005 | 0.018 | 0.009 | 0.018 | 0.006 | 0.018 | 0.006 | 0.017 |
| Fixed Effects | Country; Days since first case | | | | | | | |

Robust standard errors clustered at the country level.

* $p < 0.10$;

** $p < 0.05$;

*** $p < 0.01$.

shopping list of poor people, especially during times of economic hardships. The lack of access to adequate hand-washing facilities might also hinder their ability to combat the virus, even in the presence of greater stringency.

The idea of instilling mobility restrictions is to flatten the curve and thereby lower the disease burden on the health infrastructure. However, most less-developed countries have a limited number of hospital beds and ventilators. If these are already over-whelmed and inaccessible, flattening the curve is only marginally useful compared to countries with better and accessible health infrastructure, and the effect of stringency measures would be, accordingly, much lower. Furthermore, the higher population density in less-developed countries could

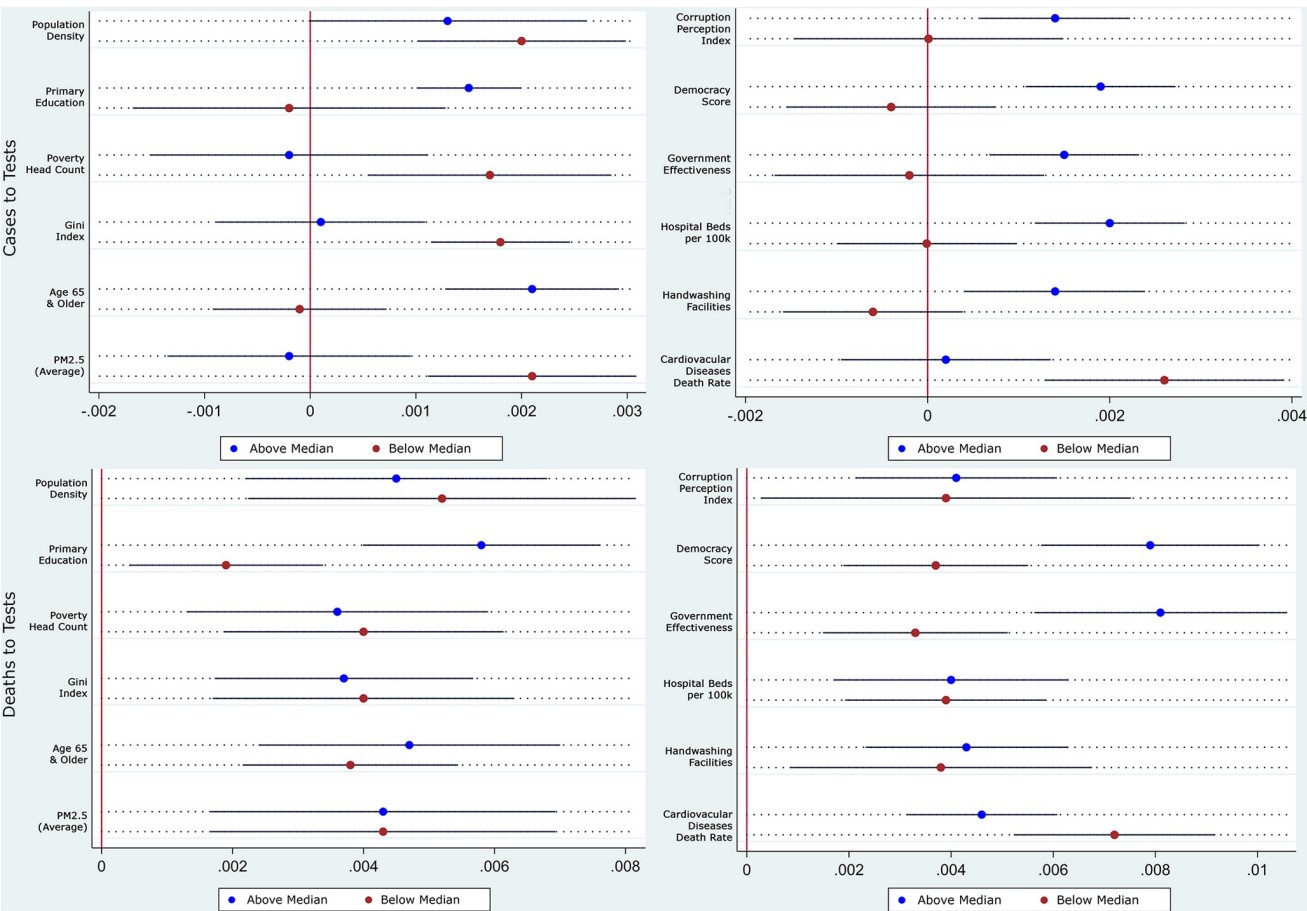

**Fig 4. Coefficient plots of hetegeneous impact of transit station mobility on growth rates: Recursive mixed-process model.** Coefficient plots are constructed using results from Table 8 with 90% confidence intervals.

mean a higher rate of human-to-human contact and transfer even with lower mobility than richer countries. Finally, another reason could be that the poor in the less-developed countries lack the knowledge of best practices to follow when a person who has tested positive is isolated either at home or at the hospital. Poorer government effectiveness and more corruption also mean sluggish enforcement of recommended best practices.

Whatever be the reason(s), one clear inference from the final set of results is that mobility measures alone were not and will not be sufficient to contain the contagion in developing countries. What is worse is that on top of the relatively worse performance of a decrease in mobility in controlling the spread, the economic cost of these restrictions is also higher in these countries. With weaker institutions, social security support, and reliance on daily wages for consumption, restrictions on economic activity mean that poorer countries face a catch-22 much worse than the richer countries. Finding a solution could be difficult without external support to implement complementary measures.

## 4 Conclusion

Some have claimed that governments across the world have responded slowly and insufficiently to the COVID-19 pandemic [55]. Others have highlighted the real threats of stricter restrictions [56]. It is, therefore, imperative to understand how effective the restrictions

implemented by the countries across the world are. Compared to earlier evaluations of these restrictions that document a strong negative association between the stringency of the restrictions and the spread of the disease, we use an instrumental variable approach to estimate the causal effect of the restrictions.

We find that while the restrictions implemented affected mobility and the spread of the disease, there was considerable heterogeneity across countries. While stricter measures reduce mobility more in less-developed countries, they do not contain the contagion as effectively as they do in developed countries. Thus, it would seem less-developed countries with weaker institutions have less to gain from stricter mobility restrictions. This could result from the lower levels of awareness, poorer health conditions and practices, and worse economic conditions in these countries. The results highlight the need to complement restriction policies with awareness, economic, and health assistance schemes.

It is, however, unclear what these complementary policies could be. From direct monetary help to only partial shutdowns, there is a range of policies to choose from. Future research should investigate the effectiveness of these alternative complementary policies in increasing the effectiveness of the mobility and activity restrictions in developing nations.

## Supporting information

**S1 Appendix. Appendix to "pandemic catch-22: The role of mobility restrictions and institutional inequalities in halting the spread of COVID-19".**
(PDF)

## Acknowledgments

We extend gratitude to Ishita Chatterjee, David Lagakos, Charles Wyplosz, Selim Raihan, Selim Jahan, Eleanor Wiseman, and participants of the "University of Exeter TARC Workshop on COVID-19 and Developing Countries" & "SANEM International Development Conference 2020 on COVID-19 and Development Challenges" for discussions and feedback on the manuscript leading to significant improvements. This research was completed while Adnan M. S. Fakir was in receipt of an Australian Government Research Training Program Scholarship at The University of Western Australia.

## Author Contributions

**Conceptualization:** Adnan M. S. Fakir, Tushar Bharati.

**Data curation:** Adnan M. S. Fakir.

**Formal analysis:** Adnan M. S. Fakir, Tushar Bharati.

**Investigation:** Adnan M. S. Fakir, Tushar Bharati.

**Methodology:** Adnan M. S. Fakir, Tushar Bharati.

**Project administration:** Adnan M. S. Fakir.

**Resources:** Adnan M. S. Fakir.

**Software:** Adnan M. S. Fakir.

**Supervision:** Tushar Bharati.

**Validation:** Adnan M. S. Fakir.

**Visualization:** Adnan M. S. Fakir.

**Writing – original draft:** Adnan M. S. Fakir, Tushar Bharati.

**Writing – review & editing:** Adnan M. S. Fakir, Tushar Bharati.

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
