## [Decision Letter · Decision Letter 0]

26 Apr 2021

PONE-D-21-02862

Pandemic Catch-22: The role of mobility restrictions and institutional inequalities in halting the spread of COVID-19

PLOS ONE

Dear Dr. Fakir,

Thank you for submitting your manuscript to PLOS ONE. After careful consideration, we feel that it has merit but does not fully meet PLOS ONE’s publication criteria as it currently stands. Therefore, we invite you to submit a revised version of the manuscript that addresses the points raised during the review process.

Manuscript need minor revision.

We look forward to receiving your revised manuscript.

Kind regards,

Faisal Abbas, PhD

Academic Editor

PLOS ONE

Additional Editor Comments:

This manuscript need minor revision before it is acceptable for publication in the journal.

Journal Requirements:

Reviewers' comments:

Reviewer's Responses to Questions

**Comments to the Author**

1. Is the manuscript technically sound, and do the data support the conclusions?

Reviewer #1: Yes

Reviewer #2: Yes

2. Has the statistical analysis been performed appropriately and rigorously? 

Reviewer #1: Yes

Reviewer #2: Yes

3. Have the authors made all data underlying the findings in their manuscript fully available?

Reviewer #1: Yes

Reviewer #2: Yes

4. Is the manuscript presented in an intelligible fashion and written in standard English?

Reviewer #1: Yes

Reviewer #2: Yes

5. Review Comments to the Author

Reviewer #1: This is an original study which aims to estimate the causal effect of mobility and activity restrictions, and the growth rate of confirmed cases and deaths attributed to COVID-19.

Overall, this is a well-designed, implemented, and written manuscript.

The authors did not describe the mobility and activity restriction characteristics according to country. This is mandatory because several countries adopted a rigid restrictions, while other countries adopted a flexible restrictions.

I recommend to move and include the sections of strength and limitations of study at the end of the discussion part.

I strongly recommend the authors seek English language revision for this manuscript. I believe this would help clarify some of the expressions and sentences that are currently not appropriate or incomprehensible.

Reviewer #2: The manuscript is a very interesting read. A comprehensive analysis is done regarding mobility restrictions and way these are impacting spread of this disease. I believe that a fairly reasonable job is done regarding the estimation and analysis of very diversified data sets.

6. PLOS authors have the option to publish the peer review history of their article (what does this mean?). If published, this will include your full peer review and any attached files.

Reviewer #1: No

Reviewer #2: No

---

## [Author Response · Author response to Decision Letter 0]

14 May 2021

Response to reviewer 1:

Reviewer #1: This is an original study which aims to estimate the causal effect of mobility and activity restrictions, and the growth rate of confirmed cases and deaths attributed to COVID-19. Overall, this is a well-designed, implemented, and written manuscript.

The authors did not describe the mobility and activity restriction characteristics according to country. This is mandatory because several countries adopted a rigid restrictions, while other countries adopted a flexible restrictions.

Authors: Thank you for raising this pertinent point. We discuss this in section 2.1.2 when describing the Oxford Stringency Index. We agree that the variation in the stringency measures needs to be carefully addressed and made transparent. In response to the reviewer’s comment, we have now included a set of event graphs that report the variation in the Stringency Index by country over time. These graphs, one for each of the 127 countries in our sample, are collected under Figures S1 to S9 in the appendix (Supporting Information). We have also incorporated the following paragraph in section 2.1.2. 

“It is also important to note while some countries enacted rigid mobility and activity restrictions, other countries adopted more flexible measures. Further, these levels of flexibility/rigidity have changed within a country over time. OxCGRT integrates these fluctuations into their stringency index by categorizing each of the nine indicators into ordinal levels by the rigidity of the restriction. For example, school closures are categorized into ``0 - no measures; 1 - recommend closing or all schools open with alterations resulting in significant differences compared to non-Covid-19 operations; 2 - require closing (only some levels or categories, for eg. just high school, or just public schools); 3 - require closing all levels'' (Hale and Webster, 2020). The final stringency index is then a composite weighted index where higher values reflect the levels of rigidity of the restrictions. Please refer to Hale and Webster (2020) for details on the index's construction. Figures S1 to S9 in the appendix provide event graphs of the stringency index by country over time for all 127 countries in our sample. Values above 50 can be interpreted as the country undertaking relatively stricter measures.”

Reviewer #1: I recommend to move and include the sections of strength and limitations of study at the end of the discussion part.

Authors: Thank you for the recommendation. We agree that including the limitations of the study at the end of the discussion is the usual structure of manuscripts. But we believe it is better to discuss the limitations to the data we use for the empirical analysis in this study before we introduce the empirical strategy. This is because the discourse on the data limitations shape and connect to the empirical strategy we employ and elaborates the bounds within which the results should be interpreted. With the present structure, the reasons behind our non-conventional specification choices are also clear to the reader. We will request to be allowed to keep the limitations section at its current position within the manuscript. 

Reviewer #1: I strongly recommend the authors seek English language revision for this manuscript. I believe this would help clarify some of the expressions and sentences that are currently not appropriate or incomprehensible.

Authors: Thank you for the recommendation. We have gone through the entire manuscript multiple times now. We have corrected grammatical errors and amended sentences to improve clarity. We also have had the manuscript professionally copy-edited. 

Response to reviewer 2:

Reviewer #2: The manuscript is a very interesting read. A comprehensive analysis is done regarding mobility restrictions and way these are impacting spread of this disease. I believe that a fairly reasonable job is done regarding the estimation and analysis of very diversified data sets.

Authors: We thank the reviewer for the encouraging remarks.

---

## [Editor Report · Decision Letter 1]

3 Jun 2021

Pandemic Catch-22: The role of mobility restrictions and institutional inequalities in halting the spread of COVID-19

PONE-D-21-02862R1

Dear Dr. Fakir,

We’re pleased to inform you that your manuscript has been judged scientifically suitable for publication and will be formally accepted for publication once it meets all outstanding technical requirements.

Kind regards,

Faisal Abbas, PhD

Academic Editor

PLOS ONE

Additional Editor Comments (optional):

Accept.
---

## [Editor Report · Acceptance letter]

17 Jun 2021

PONE-D-21-02862R1 

Pandemic Catch-22: The role of mobility restrictions and institutional inequalities in halting the spread of COVID-19 

Dear Dr. Fakir:

I'm pleased to inform you that your manuscript has been deemed suitable for publication in PLOS ONE. Congratulations! Your manuscript is now with our production department. 

Kind regards, 

on behalf of

Dr. Faisal Abbas 

Academic Editor

PLOS ONE